15

# **Evaluation of nine gridded daily weather reconstructions for the European heatwave summer of 1807**

Peter Stucki<sup>1,2</sup>, Stefan Brönnimann<sup>1,2</sup>, Noemi Imfeld<sup>1,2</sup>, Lucas Pfister<sup>1,2</sup>, Conall E. Ruth<sup>1,2</sup>, Yannis Schmutz<sup>1,2,3</sup>, Yuri Brugnara<sup>1,2</sup>, Martin Wegmann<sup>1,2</sup>, Rajmund Przybylak<sup>4,5</sup>, Janusz Filipiak<sup>6</sup>

- 5 1 Oeschger Centre for Climate Change Research, University of Bern, Bern, 3012, Switzerland
  - <sup>2</sup> Institute of Geography, University of Bern, Bern, 3012, Switzerland
  - <sup>3</sup> Berner Fachhochschule Technik und Informatik, Bern, 3012, Switzerland
  - <sup>4</sup> Faculty of Earth Sciences and Spatial Management, Nicolaus Copernicus University, Toruń, Poland
  - <sup>5</sup> Centre for Climate Change Research, Nicolaus Copernicus University, Toruń, Poland
- Operation of Physical Oceanography and Climate Research, Faculty of Oceanography and Geography, University of Gdańsk, Gdańsk, Poland

Correspondence to: Peter Stucki (peter.stucki@unibe.ch)

Abstract. Recent research of early instrumental measurements combined with numerical-statistical techniques has contributed to global atmospheric reanalysis as well as regional products that cover pre-1850 weather. The advent of machine learning (ML) raises the question of how well we can reconstruct weather from the distant past using both established and emerging approaches. Here, we evaluate nine such approaches to reproduce the daily weather during Europe's hot summer of 1807. The datasets examined include the Twentieth Century Reanalysis (20CR) and enhanced versions (via additional assimilation, dynamical downscaling), an analog resampling product, as well as ML reconstructions that use neural networks (along with video-inpainting methods or variational auto-encoders). Validation is based on early station measurements, documentary information, statistical diagnostics, and a semi-quantitative assessment of atmospheric flow.

We find that the summer of 1807 can be considered a prototype, pre-industrial heatwave summer, with three extremely hot episodes and maximum temperatures exceeding 30 – 35 °C in Central Europe. Most approaches achieve mean correlations (anomalies form the seasonal cycle) above 0.75 for temperature and centered Root Mean Square Error values below 3 °C, though variability tends to be underestimated. This speaks for overall robust reconstructions given the distant past and scarce underlying weather information. Skill scores for almost all reconstructions indicate that they are reliable in discriminating very hot from cooler (high-pressure from lower-pressure) conditions. Improved spatial skill with respect to 20CR for stations in Central and Northeastern Europe can be attributed to the increased influence of newly ingested weather information on the atmospheric reconstructions.

The atmospheric flow-aware approaches reproduce plausible large-scale features such as ridges of high pressure and associated belts of hot air, whereas data-driven ML approaches excel statistically in replicating station variability but often produce less realistic circulation patterns. The analog method yields balanced but less intense reconstructions, and the high-resolution dataset aligns best with heat intensities in the Alpine region.

Such trade-offs leave users choose between computational efficiency, statistical performance, and physically coherent circulation. Future developments need to address uncertainties in the early measurements. In turn, the analyses also emphasize the value of high-quality early weather records to produce and validate gridded reconstructions.

#### Introduction

- Over the last decades, summer heatwaves across the globe have become more frequent, intense, and prolonged (e.g. Perkins-Kirkpatrick and Lewis, 2020, and references therein), and the IPCC projections assume even greater changes in the future (IPCC, 2023). Typically, such changes or deviations are calculated with respect to a climatological baseline. The IPCC reports use the period of 1850 to 1900 (Schurer et al., 2017), as 'an approximation for pre-industrial' conditions.
- No encompassing comparisons to pre-1850 conditions have been made in the IPCC reports so far. The IPCC justifies this with insufficient and incomplete observations to estimate climate variables like global surface temperatures before around '1850 1900' (e.g. Hawkins et al., 2017). In recent years, however, there have been substantial efforts to research historical weather records. For instance, a vast number of early (here pre-1850) instrumental measurements were searched and processed in digitization and homogenization efforts (Allan et al., 2011; Auer et al., 2007; Brönnimann et al., 2019; Brunet and Jones, 2011; Cram et al., 2015; Slonosky and Sieber, 2020) and facilitated by platforms like the WMO Data Rescue initiatives (WMO, 2020), the Copernicus C3S Data Rescue Service (Copernicus, 2025), or the Southern Weather Discovery campaign (Lorrey et al., 2022).
  - Vital steps towards 'completing the information for pre-industrial periods' have been made by producing global to regional reconstructions of climate variables. For instance, the HISTALP dataset (Auer et al., 2007) provides a spatially interpolated field of absolute monthly temperature values for the greater Alpine region.
- Another vital step has been the development of four-dimensional, gridded, numerical, and typically global weather datasets, known as reanalyses. Reanalyses use data assimilation (of weather observations) to fit numerical weather-model output to available observational data, considering potential errors of air pressure or temperature, among others. An example is the Twentieth Century Reanalysis (Compo et al., 2006, 2011; Slivinski et al., 2019), which assimilates pressure measurements from the International Surface Pressure Databank (Compo et al., 2019; Cram et al., 2015). The current version 3 (20CR hereafter) extends back to 1806.
  - More recently, augmented 20CR versions, i.e. with new observations, temperature as an additional variable, and improved assimilation, have been presented and tested, such as for windstorm Ulysses on Great Britain in 1903 (Hawkins et al., 2023) or using temperature, pressure, geopotential height, and total ozone for the case years 1807, 1877 / 1878, and 1926 / 1927 over Europe in an off-line assimilation (Brönnimann, 2022). Furthermore, dynamical downscaling of global reanalyses has been applied to achieve higher spatial and temporal resolutions (Gómez-Navarro et al., 2018; Michaelis and Lackmann, 2013). For Central Europe, gridded daily fields (e.g., of temperature and precipitation) have been produced using the analog resampling method (ARM), which compares atmospheric patterns from historical station data with a pool of analogs in modern gridded


datasets (Flückiger et al., 2017; Rössler and Brönnimann, 2018). The same way as for global reanalyses, additional observations can be assimilated to improve the reconstruction (Imfeld et al., 2023; Pfister et al., 2020).

In addition to such refinements of physics-constrained or flow-aware approaches, rapid progress has also been made in the implementation of machine-learning (ML) algorithms (e.g. Bonavita et al., 2021), where the training of ML models is done in recent decades and the gained knowledge applied to historical data for weather and climate reconstruction. For instance, Schmutz et al. (2024) presented a three-dimensional convolutional neural network inspired by the task of video inpainting for reconstructing gridded daily fields of temperature and pressure across Europe. Wegmann and Jaume-Santero (2023) investigated long short-term memory models to reconstruct global temperature anomalies on a monthly resolution for 400 years, although their approach has not yet been tested for daily weather. Other deep-learning architectures have also been proposed for such applications, including variational auto-encoders (Brohan, 2022; Slivinski et al., 2025).

To date, we hence have a variety of approaches, that allow us to create 'complete', that is four-dimensional, continuous datasets reaching back in time for centuries, and as a consequence, pre-industrial periods such as the turn of the 18<sup>th</sup> to the 19<sup>th</sup> century come into the spotlight of numerical weather analyses. This motivates us to assess the capabilities and limitations of a number of such numerical approaches in reproducing the daily weather during a pre-1850 hot summer.

Concretely, we analyze the hot summer of 1807 in Europe, with a particular focus on the Alpine region. In HISTALP, the month of July in 1807 appears as the hottest in the area when compared to a pre-1850 reference period of 1806 – 1835. According to a more recent temperature reconstruction (Rohde and Hausfather, 2020), the summer of 1807 ranks among the ten warmest European summers between 1750 and 2000. Availability of documentary and instrumental sources, particularly early meteorological measurements, is relatively good for this region and period. Evidence for an extended hot summer 1807 across Central Europe has been found in a range of proxy series and descriptive sources, e.g. from compilations such as EuroClimHist (Rohr, 2016) or tambora.org (Riemann et al., 2015). For Germany, the latter has 1530 entries of temperature indices for June, July, and August since 1500 A.D., and August 1807 was among the hottest 2 %, i.e. with the highest index. More anecdotal reports about the heat in summer 1807 include messages to and from Alexander von Humboldt (Erdmann and Brönnimann, 2023): There are complaints about 'intense and long-lasting heat' episodes from July to the end of August with

temperatures of around 32 °C in Fulda, Germany, and 'unbearable heat' in August in Göttingen, Germany. Humboldt also noted a daily maximum temperature of around 37 °C on 14 July 1807 in Rome, and around 29 °C on 23 August 1807 in Göttingen.

The article is organized as follows: In Section 2, we inform about the observations and measurements used, as well as the gridded reconstruction approaches. In Section 3, we describe the statistical diagnostics used for the evaluation. In Section 4, we first characterize the hot summer 1807 in Europe from weather records and 20CR. In Section 5, we use nine approaches for a contest of reconstructing temporal as well as spatial aspects of the hot summer. The competing approaches are the 20CR ensemble mean and three enhanced derivates from 20CR, a dynamical downscaling product, an analog method product, and three ML-based approaches. With a range of statistical diagnostics, we assess the similarity and discrimination (of classes) of reconstructed vs observed daily temperature and pressure during the summer (half year) of 1807 as well as the spatial

representation of temperature and pressure for three hot episodes. In a semi-quantitative approach, we also assess the quality of the underlying weather observations. In Section 6, we provide a summary and a conclusion.

#### 2 Data




## 2.1 Databases and collections of documentary and instrumental data

Documentary and instrumental data used in this article have been found in specific sources and databases for historical climatology (Erdmann and Brönnimann, 2023; Hari, 2021; Riemann et al., 2015; Rohr, 2016). For Europe in the period around 1807, weather diaries were more and more completed with records from early instrumental measurements. Single measurement series or small collections of such series have been made available in a range of research and digitization efforts, such as the series from Geneva by Auchmann et al. (2012) or the series from Vilnius, Wroclaw, and Warsaw by Rajmund Przybylak (pers. comm.), and Gdansk by Janusz Filipiak (pers. comm.). More series of daily resolved, early instrumental measurements were researched, processed and published by Brönnimann et al. (2019), Brugnara et al. (2015, 2016, 2020, 2022), and Pfister et al. (2019). All these series were used as references, or 'ground truth', for the reconstruction approaches (**Table 1**; see also **Figure S1** in the Supplement). Note that all maps hereafter display modern national borders rather than the contemporary borders of 1807. Here, we focus on observations of temperature and pressure on a daily to sub-daily basis. From these stations, five were set aside for validation to cover areas with different densities of stations (Basel, Zurich Feer, Bologna, Gdansk, and Vienna). The records of the stations Aarau, Bern, Marschlins, Delémont, Vevey and Delft are not used for assimilation and ingestion, but are also not treated as independent in order to reduce the weight of many neighboring stations in the area.

# 120 Table 1: Stations used for the assimilation experiments 1807.\*

| Station            | Var     | Time        | Reference                        | CRM | CRx   | non-CR |
|--------------------|---------|-------------|----------------------------------|-----|-------|--------|
| Armagh             | р       | 3x daily    | ISPD p                           |     |       | р      |
| Aarau              | ta, p   | daily       | Brugnara et al., 2020            |     |       |        |
| Barcelona          | ta      | daily       | Rodríguez et al., 2001 ta        |     |       | ta     |
| Basel              | ta, p   | daily       | Brugnara and Brönnimann, 2023    | val | val   | val    |
|                    |         |             | PALAEO-RA                        |     |       | ta, p  |
| Berlin             | ta, p   | 3x daily    | (Brönnimann et al., 2020)        |     |       |        |
| Bern (obs. Studer) | ta, p   | daily       | Brugnara et al., 2020            |     |       |        |
| Bologna            | ta      | daily       | Camuffo et al., 2017 val         |     | val   | val    |
| Cadiz              | ta      | daily, noon | Barriendos et al., 2002          |     | ta    | ta     |
| Central Belgium T  | ta      | daily, noon | Demarée et al., 2002 ta          |     | ta    | ta     |
| Central England T  | ta      | daily       | Parker et al., 1992 ta           |     | ta    | ta     |
| Delémont           | ta, p   | daily       | Brugnara et al., 2020            |     |       |        |
|                    |         |             | KNMI (Royal Netherlands          |     |       |        |
| Delft              | ta      | daily       | Meteorological Institute)        |     |       |        |
| Gdansk             | ta, p   | daily       | Janusz Filipiak pers.comm.       | val | val   | val    |
| Geneva             | ta, p   | 14, daily   | Auchmann et al., 2012; ISPD      | р   | ta    | ta, p  |
| Haarlem            | ta, p   | daily       | KNMI; ISPD                       |     |       | ta, p  |
| Hohenpeissenberg   | ta, p   | daily       | Winkler, 2006; ISPD              | p   | ta    | ta, p  |
| Karlsruhe          | р       | daily       | Kunz et al., 2022                |     | р     | р      |
| London             | р       | daily       | Cornes et al., 2012              |     | р     | р      |
| Marschlins         | ta, p   | daily       | Brugnara et al., 2020            |     |       |        |
| Milano             | ta      | daily       | Maugeri et al., 2002             |     | ta    | ta     |
| Mulhouse           | ta, p   | daily       | Brugnara et al., 2020 ; ISPD     | р   | ta    | ta, p  |
| Padova             | ta, p   | daily, noon | Camuffo, 2002                    |     | ta, p | ta, p  |
| Paris              | р       | daily       | Cornes et al., 2012b             | р   |       | р      |
| Paris              | ta      | 12          | Daniel Rousseau, pers. comm.     |     | ta    | ta     |
| Prague             | ta      | daily       | Stepanek, 2005                   |     | ta    | ta     |
| Rovereto           | ta, p   | 8 or 16     | Brugnara et al., 2022            |     | ta, p | ta, p  |
| Schaffhausen       | ta, p   | 3x daily    | Brugnara et al., 2020            |     | ta, p | ta, p  |
| St. Petersburg     | ta, p   | daily       | Jones and Lister, 2002           |     | ta, p | ta, p  |
| Stockholm          | ta, p   | daily       | Moberg et al., 2002; ISPD        | р   | ta    | ta, p  |
|                    |         | •           | Di Napoli and Mercalli, 2008;    | p   | ta    | ta, p  |
| Torino             | ta, p   | daily       | ISPD                             |     |       | •      |
| Uppsala            | ta      | daily       | Moberg et al., 2002              |     | ta    | ta     |
| Valencia           | ta, p   | 13          | Domínguez-Castro et al., 2014    |     | ta, p | ta, p  |
| Vevey              | ta, p   | daily       | Brugnara et al., 2020            |     |       |        |
| Vienna             | ta      | 3x daily    | Geosphere Austria                | val | val   | val    |
| Vilnius            | ta      | daily       | Rajmund Przybylak pers.comm.     |     |       | ta     |
| Warsaw             | ta      | daily       | Rajmund Przybylak pers.comm.     |     |       | ta     |
| Wroclaw            | ta      | daily       | Rajmund Przybylak pers.comm.     |     |       | ta     |
| Yilitornio         | ta, p   | 14, daily   | Klingbjer and Moberg, 2003; ISPD | р   | ta    | ta, p  |
| Žitenice           | ta. p   | 14          | Brázdil et al., 2007             | -   | ta, p | ta, p  |
| Zwanenburg         | ta      | 3x daily    | KNMI                             |     | ta    | (ta)   |
| Zürich (obs. Feer) | ta, p   | daily       | Brugnara et al., 2022a           | val | val   | vaĺ    |
|                    | ··· / F |             | <u> </u>                         |     |       |        |

<sup>\*</sup> Abbreviations are: Var for variable, ta for air temperature, p for pressure, obs for observer, CRx relates to the three 20CR-derived approaches (see Table 1 in Brönnimann 2022 for details on these stations), and non-CR relates to all other datasets according to Table 2. The CRx datasets use the values close to noon time whereas the non-CR datasets use daily values where

available. Note that WRF does not assimilate. Stations used for validation (val) are in italics, stations with no indicated use are only shown in the analyses but not treated as independent. Zwanenburg is used for ta by ARM only, instead of Haarlem.

## 2.2 Numerical, gridded, observation-based atmospheric datasets

A number of gridded, numerical, and typically global weather datasets have been developed by means of sophisticated regionalization approaches and based on series of (early) instrumental measurements similar to the ones mentioned above.

**Table 2** summarizes the approaches used for comparisons in this study.

Table 2: Numerical, gridded datasets used for reconstructions of the summer 1807 in Europe.\*

| Dataset                                        | Abbr. | Additional obs. | Horizontal resolution | Horizontal extent for this study                           | orTemporal<br>resolution | Temporal extent            |
|------------------------------------------------|-------|-----------------|-----------------------|------------------------------------------------------------|--------------------------|----------------------------|
| Twentieth Century<br>Reanalysis 20CR           | CRM   |                 |                       | 19.688 W – 40.078 E<br>29.825 N – 69.825 N                 | 3-hourly                 | 1806 to 2015               |
| 20CR_plus                                      | CRP   | yes             | . 1°                  |                                                            | daily                    | 1807                       |
| 20CR best member                               | CRB   | no              |                       |                                                            |                          |                            |
| 20CR_plus best member                          | СРВ   | yes             |                       |                                                            |                          |                            |
| Weather Research and Forecast Model            | WRF   | no              | 3 km                  | 0.995 E – 18.624 E<br>41.799 N – 52.699 N<br>(curvilinear) | 6-hourly                 | 1806-12-01 –<br>1807-10-12 |
| Analog Resampling and data assimilation Method | ARM   | yes             | 0.25°                 | 20 W – 40 E<br>30 (ta), 35 (p) N – 70 N                    | daily                    | 1807                       |
| Three-Dimensional Convolutional Neural Network | TNN   | yes             | 1°                    | 21.625 W - 41.375 E<br>35.625 N - 66.625 N                 | daily                    | 1807                       |
| One Dimensional Convolutional Neural Network   | ONN   | yes             | 0.25°                 | 22 W – 41 E<br>36 N – 67 N                                 | daily                    | 1807 (w/o<br>1807-01-01)   |
| Variational Auto-Encoder                       | VAE   | yes             | 1°                    | 22 W – 41 E<br>36 N – 67 N                                 | daily                    | 1807                       |

<sup>\*</sup>Additional obs. stands for ingestion and assimilation of more station observations than in 20CR.

# 2.2.1 Twentieth Century Reanalysis 20CR

For synoptic analyses on a (sub-) daily scale, the NOAA / CIRES / DOE 20th Century Reanalysis version 3 (20CR) is available from the NOAA PSL (Slivinski et al., 2019; Compo et al., 2011). 20CR is also used for initial and boundary conditions for the




dynamical downscaling experiments. Version 3 comes with a 1° x 1° horizontal global grid (approx. 75 km over Europe), 28 vertical pressure levels, and a 3-hourly temporal resolution going back to 1836. Only surface pressure observations are assimilated (see **Table 1**). For our study, we use the experimental extension with 80 members reaching back to 1806, available (at the time of writing) from the US National Energy Research Scientific Computing Center NERSC web portal at https://portal.nersc.gov/project/20C\_Reanalysis/. Over the time period of our 1807 case, three stations in the greater Alpine region are assimilated in 20CR (Geneva, Turin, and Hohenpeissenberg). Here, we typically show the 20CR ensemble mean (CRM).

## 2.2.2 20CR plus, 20CR best member, 20CR plus best member

20CR\_plus (CRP) is obtained by assimilating observations into the 80-member ensemble of 20CR using an offline Ensemble Kalman Filter approach (see Brönnimann, 2022). In this approach, all ensemble members are corrected towards the observations each day but the model is not rerun. In contrast to 20CR, 20CR\_plus also assimilates temperature observations (see **Table 2**), although not all stations from **Table 1** were available at the time of production. Furthermore, if several stations fell within the same 20CR grid cell (e.g., stations in the Netherlands or Switzerland), only one was kept. As there might be systematic differences between 20CR and observations, both pressure and temperature observations are adjusted to the corresponding 20CR data for the year 1807 before assimilation. For temperature, this is done by fitting a seasonal cycle to each dataset based on the first two harmonics and subtracting the difference. For the pressure series, the overall mean difference is subtracted. The observation error is assumed to be 3^2 K and 3^2 hPa for temperature and pressure, respectively. We then use a sequential implementation of the Ensemble Kalman Filter, with no localization of the background error covariance matrix. Observations are not assimilated if their departure from 20CR is larger than 3 times the standard error, which is defined as the square root of the sum of the error variances of 20CR and observations. Leave-one out cross validation shows a significant improvement over 20CR (Brönnimann, 2022).

For the "best member" products, i.e. 20CR best member (CRB) and 20CR\_plus best member (CPB), we determined for each day the member of 20CR that fits best with all observations to be assimilated based on the Eucledian distance of the standardized anomalies. CRB and CPB are concatenations of this best member for every day before (CRB) and after (CPB) assimilation of the observations.

## 2.2.3 Regional weather model WRF

We use the non-hydrostatic Advanced-Research Weather Research and Forecast Model Version 4.3.3 (WRF-ARW; hereinafter WRF; Skamarock et al., 2021) for dynamical downscaling from the 20CR ensemble mean with two domains (15 and 3 km spatial resolution) and 6-hourly resolution. We use the Thompson microphysics scheme, the Yonsei University (YSU) scheme for the planetary boundary layer together with the revised MM5 surface layer scheme, the scale-aware Grell-Freitas cumulus





scheme and the RRTMG radiation schemes. The NoahMP model is used for the land surface. Please refer to the NCAR Technical Note on WRF 4.3.3 (Skamarock et al., 2021) for references to the specific schemes. Spectral nudging (corresponding to a wavelength of about 1000 km) is applied to temperature, wind, and geopotential fields above the planetary boundary layer in the outer 15-km domain. The WRF model is initialized on 1806-12-01 at 00 UTC, allowing for several weeks of model spin-up, and the model output is saved at 6-hourly resolution. WRF relies on the 20CR input only, that is, no stations from **Table 1** are assimilated.

## 2.2.4 Analog resampling and data assimilation

The analog resampling and data assimilation method (ARM) is based on the assumption that over time, similar atmospheric states occur repeatedly, and thus similar surface fields of pressure and temperature, for example. Therefore, a historical target day can be reconstructed by first searching for the most similar days in a reference period and then resampling the required fields based on these analog days. Here, we use sea-level pressure and 2-meter temperature fields with a 0.25 ° horizontal resolution from the ERA-5 reanalysis (Hersbach et al., 2020) for the period from 1950 to 2020 as reference fields. As reference stations, we use temperature and pressure observations from ECA&D (Klein Tank et al., 2002) and MeteoSwiss (Begert et al., 2007). Before resampling, an offset is subtracted from the temperature data to account for the inter-centennial climate change. This offset is the difference in zonal mean land-only temperature from the EKF400 palaeo-reanalysis (Valler et al., 2021) between the two periods. The best 50 analog days between the historical and the reference period are determined using the root mean square error (RMSE) from deseasonalized, detrended, and standardized observations. The same observations are assimilated onto the resampled fields using an ensemble Kalman filter (EnKF; Bhend et al., 2012; Franke et al., 2017). The observation errors are calculated from a linear regression between the gridded and station data in the reference period. Observation errors are calculated from a linear regression between the daily differences in variance and the spatial distance of every station pair (Wartenburger et al., 2013). Furthermore, a spatial localization is performed for cut-off distances of 1500 km for temperature and 2000 km for sea-level pressure. More details can be found in Pappert et al. (2022).

#### 2.2.5 Three-Dimensional Convolutional Neural Network

The Three-Dimensional Convolutional Neural Network WeRec3D (Schmutz et al., 2024; abbreviated TNN in this study) is a deep learning architecture tailored to weather reconstruction. Inspired by video inpainting, it is based on a three-dimensional convolutional neural network, which allows to model space and time dimensions simultaneously. In contrast to traditional meteorological techniques that operate on anomalies (Qian et al., 2021), TNN shows superior results when modeling climatology directly. It further leverages five custom techniques that enhance the algorithm specifically for weather reconstruction. TNN was trained and validated in a self-supervised manner using ERA5's daily sea-level pressure and 2-meter temperature fields with a 1° x 1° horizontal grid over Europe. On a hold-out set from 1950 to 1954, the validation results in an







MAE of 1.11 °C and 1.99 hPa. The authors further tested its reconstruction capability on the heat wave of 1807 using a leaveone-out validation in space. Compared to the historical measurements, the reconstructed time series exhibit a correlation of at least 0.91, with a maximum normalized RMSE and standard deviation delta of 0.58 and 0.51, respectively.

## 2.2.6 Long-short Term Memory models and One-Dimensional Convolutional Neural Networks

Using the same training system as TNN and ARM, we investigate out-of-the-box Long-Short Term Memory (Hochreiter and Schmidhuber, 1997) and One-Dimensional Convolutional Neural Networks (ONN hereafter; Kiranyaz et al., 2021) for the regression task of reconstructing 2m-temperature and sea level pressure on the 0.25 ° ERA5 grid. For this task, the 2D spatial field was initially reshaped into a one-dimensional vector, and eventually reshaped to a 2D field. The best from a variety of dropout and layer architectures is used for the final reconstruction. We use the ADAM optimizer (Kingma and Ba, 2015) with a learning rate of 10<sup>-4</sup>, a batch size of 128, a hyperbolic tangent activation function and the Mean Squared Error as loss function. We then store the model weights with the lowest validation loss after not improving over 20 epochs. A more detailed discussion of this approach can be found in Wegmann and Jaume-Santero (2023). After removing February 29th entries, training sample size for this task is N=21900. 80% of this data is used for training, and 20% is used for validation. We find no overfitting in this training sample size, and as such no dropout is implemented. To account for missing data in the 1807 reconstruction, we create a mask of missing values from each individual station in 1807 and then apply that mask for each year and each individual station in the training period 1960 - 2020. We investigate different amounts of features in the reconstruction, from simply using one feature (e.g. temperature only) to using a maximum of eight features (longitude, latitude, altitude, temperature, pressure, pressure tendency, weather type, day of year). Here we focus on results with seven (no day-of-year information) or eight features.

## 2.2.7 Variational Auto-Encoder

We implement a novel machine learning model based on a variational autoencoder (VAE) architecture. While the use of such a model for reconstructing historical weather fields has previously been proposed (Brohan, 2022), to our knowledge no such studies have been published. The VAE comprises a pair of 2D convolutional neural networks in an encoder-decoder arrangement. During training, the encoder reduces the complete fields of temperature and pressure for a given day to a simplified 256-dimensional representation within the model's latent space, from which the decoder then attempts to recreate the original fields. Regularizing the latent space through the inclusion of a Kullback–Leibler loss term ensures that, once trained, the decoder should be capable of generating realistic output fields from any latent representation that is provided. To apply the model, we therefore discard the trained encoder, randomly sample the latent space, and then iteratively adjust this sample up to 500 times using gradient descent to minimize the error between the resulting output of the decoder and any available observations from a given day of interest. The final iteration of this output then represents the reconstruction for that

day. To maximize the consistency with TNN, the VAE model was developed using ERA5 data for the period 1950–2020 and applied using the same set of input stations. Temperature is handled as normalized intra-monthly anomalies, created by subtracting the observed calendar-monthly means and dividing by the resulting standard deviation of the training period (1969–2020). Pressure is handled as z-normalized anomalies purely with respect to the training period. The reconstructed values are then reconverted to absolute terms using the same training-period statistics and using ModE-RA (Valler et al., 2024) for the calendar monthly means of temperature. The VAE's exact architecture and hyperparameters have been tuned to minimize reconstruction error assuming the observation availability of 1807. The evaluation on a test set (1950–1954) yields overall RMSEs of 2.52 K and 5.18 hPa. A full description of the VAE model is provided in the Supplement.

#### 235 3 Verification measures and scores

For our purposes, we use a range of verification measures and scores. The general similarity between the numerical output and the observed values (e.g. 2-meter air temperature at the station and at the nearest grid point in 20CR) over a certain period (e.g. daily values from May to September) is assessed by the three statistics that can be visualized in Taylor diagrams (Taylor, 2001a). These are the (Pearson) correlation coefficient *COR* (Eq. 1)

$$COR = \frac{\sum (x_i - \overline{x})(y_i - \overline{y})}{\sqrt{\sum (x_i - \overline{x})^2 \sum (y_i - \overline{y})^2}} \tag{1}$$

to assess linear association (irrespective of bias) between predicted and observed values, the ratio of standard deviations SDR (Eq. 2 shows SD for observed values x)

$$SD = \sqrt{\frac{1}{n-1} \sum_{i=1}^{n} (x_i - \overline{x})^2} \text{ then } SDR = \frac{SD_{simulation}}{SD_{observation}}$$
 (2)

to assess the amplitude of variability in the predicted vs. the observed series, and the centered root mean squared error *cRMSE*, where the relation to the root mean squared error *RMSE* (Eq. 3)

$$RMSE = \sqrt{\frac{1}{n-1} \sum_{i=1}^{n} (y_i - x_i)^2}$$
 (3)

is given by Eq. 4


$$(cRMSE)^2 = RMSE^2 - (\overline{y} - \overline{x})^2 \tag{4}$$

(cf. Paul et al., 2023). The cRMSE cannot be equated with the RMSE, but is mathematically equal to the standard deviation of the model error (predicted minus observed), thus gives a hint about how strongly the 'errors' in the predicted series fluctuate (Elvidge et al., 2014).

Second, we use three statistics to assess the specific ability of reconstructing more rare events, e.g. the heat-wave episodes in Europe during the summer of 1807. For this, we divide each series into three classes, which are 'hot' (>=  $90^{th}$  percentile), 'warm' ( $90^{th} > x >= 75^{th}$  percentiles), and 'cooler' (

(beyond 80<sup>th</sup> or 20<sup>th</sup> percentile, respectively) versus 'rather normal' values in-between. The use of percentiles provides information on whether high, low or normal predictions are reflected in the observations, regardless of absolute values. It also means that we have ordinal categories and imbalanced classes, but their size is the same in each sample.

The first score is the Balanced Accuracy (*BACC*), where the sample weights  $\omega$  are normalized for each class (Eq. 5; Lang et al., 2024)

$$\widehat{\omega}_i = \frac{\omega_i}{\sum_j \mathbf{1}(y_j = y_j)\omega_i} \tag{5}$$

and then Eq. 6

$$BACC = \frac{1}{\sum_{i} \widehat{\omega}_{i}} \sum_{j} 1(\widehat{y}_{i} = y_{i}) \widehat{\omega}_{i}$$
 (6)

computes a sample-weighted accuracy with scores between 0 for no accuracy and 1 for a perfect prediction with respect to all classes.

For the second score, we consider the Multi-Class Brier Score (MBR) in Eq. 7 (Lang et al., 2024),

$$MBR = \frac{1}{n} \sum_{i=1}^{n} \sum_{j=1}^{r} (I_{ij} - p_{ij})^2$$
 (7)

with r labels and where  $I_{ij}$  is set to 1 if the observation i has the true label j, and set to 0 in all other cases. The MBR assesses predicted probabilities for each class ( [0,1] in our case) versus the actual occurrence of a multi-class event (value 0 or 1). Note that the MBR does not account for sample weights, and that the range of the 3 x 3 MBR goes from 0 to 2 for best performance.

Other than that, visual tests for our results look similar from MBR and Log Loss Score (not shown).

Finally, we calculate the Gerrity Skill Score (GSS), which uses a scoring matrix  $s_{ij}$  with the values i = 1, ..., 3 depending whether a reward or penalty for each prediction / observation outcome will be assigned (Goncalves, 2023). Hence, Eq. 8

$$GSS = \sum_{i=1}^{3} \sum_{j=1}^{3} p_{ij} \ s_{ij}$$
 (8)

assesses the skill of a simulation in discriminating the classes, that is, it assesses how accurately the reconstruction predicts the correct category, relative to random predictions (Joliffe and Stephenson, 2012). The underlying Gerrity Score evaluates whether a predicted category of pressure or temperature actually occurred. In particular, the Gerrity Score strongly rewards simulations that correctly predict the less likely class, while it does much less reward conservative predictions. Smaller errors (i.e. from predicting a neighboring class) are less penalized than larger errors (from predicting more distant classes). The measure of the Gerrity Score ranges from -1 to 1, where 0 means that the prediction has no skill. The *GSS* is so called equitable, meaning that random and constant predictions yield a score of 0. A score of 1 means that all values were correctly categorized. A score of 0.5 means that the prediction is about twice as accurate as random predictions.

Then, we apply the Taylor Skill Score (TSS; Taylor, 2001b). It allows to summarize the three Taylor statistics in just one score, with low skill near 0 and maximum skill close to 1. The score rewards high correlations as well as high agreement of variance between simulated and observed values, (cf. Veiga and Yuan, 2024), such as Eq. 9

$$TSS = \frac{4*(1+COR)^4}{(SDR + \frac{1}{SDR})^2 * 2^4} \tag{9}$$

when considering the maximum COR == 1.

Finally, we use the Terrain Ruggedness Index *TRI* by Riley et al. (1999) that was originally created to quantify topographic heterogeneity in Eq. 10 as

$$TRI = \Upsilon \left[ \sum (x_{ij} - x_{00})^2 \right]^{0.5} \tag{10}$$

where  $x_{ij}$  is the neighboring grid point to the reference g-rid point  $x_{00}$ . For this study, it serves as a relative measure of 'ruggedness' of the simulated pressure and temperature fields among the numerical datasets. Thus, we use the *TRI* as implemented in the spatialEco R package (Evans and Murphy, 2023), and simply replace the elevation variable with temperature (pressure) values. To account for differences in spatial representation among the numerical datasets, we use a selected field of 13 x 7 grid points in 20CR (every 5<sup>th</sup> in west-east and south-north direction) covering roughly the area from 15W to 30E and 35N to 60N, and we use the relative nearest neighbors in the other numerical datasets. Still, we compare the TRI across models with varying spatial resolutions (most at 1°, while ARM and ONN use 0.25°), so we would expect the finer-resolution models to appear more rugged in tendency.

## 4 The summer 1807 in Europe in weather records and 20CR

In a first analysis, we step from anecdotal, descriptive evidence of "unbearable and long-lasting heat" in (Central) Europe in August 1807 (e.g. Erdmann and Brönnimann, 2023; see Section 1) to observation-driven evidence of the daily weather, i.e. we analyze information that comes from single instrumental as well as observational time series. A very helpful source to look at the weather conditions in the Swiss midlands are eye observations by Studer in the city of Bern. **Figure 1** (cf. Hari, 2021) visualizes these weather records for the year 1807.




Figure 1: Categorized daily weather records from Bern (observer Studer; data from Hari, 2021) for the year 1807.

Although a substantial number of days in the study period are missing, there were episodes with hardly any cloudiness or precipitation reported, but permanently sunny conditions. These episodes start in May, and more episodes occur in July and August. In comparison, **Figure 2** represents instrumental measurements of daily temperature at three selected locations in Europe. At the Prague station (**Figure 2a**), a first heatwave episode with no values below the long-term 95th percentile (peaks of around of 27 °C) appears in mid-July, followed by more such episodes in August, e.g. at the beginning of the month, and the hot summer concludes in an episode of more than 10 days towards the end of the month. At the Stockholm station (**Figure 2b**), the episode around 1 August is outstanding, while July and end-of-August episodes are less prominent. Also, the Zurich station (**Figure 2c and d**) exhibits the highest mean daily temperature of 27 °C on 13 July, an early-August peak, and a heatwave of 10 consecutive days with daily mean temperature above 22 °C from 20 August, a unique feature in the series.

13

Figure 2: Daily mean temperatures at Prague, Stockholm, and Zurich from May to October 1807 (black circles), and temperature quartiles wrt 1806-1835 (grey lines). Red dots mark daily mean temperature above the 95<sup>th</sup> percentiles (red horizontal line) in panels a), b), and c). Dots in panel d) indicate the maximum number of consecutive days with mean temperatures >=22 °C in the Zurich measurements. Series of 5 days or more are labeled.

Hence, we find three particular peak episodes of hot days in Central and Northern Europe from these series: These are the single hot day of 13 July 1807, three particularly hot days from 1 to 3 August 1807 as well as five days from 26 to 30 August 1807.

For these days, **Figure 3** shows temperature values near 30 °C at stations just north and south of the Alps for all three episodes (with questionable exceptions, e.g. at Marschlins and Hohenpeissenberg; cf. Section 5.3). Lower values appear over Scandinavia (between 11 and 17 °C), and near the Channel (up to around 20 °C). The synoptic weather situation becomes more apparent when mapping the respective warmest members in 20CR (**Figure 4**).





Figure 3: Analyses of daily air temperature (absolute values in the top row, anomalies wrt the reference period 1806 - 1835 in the bottom row; °C) from early instrumental measurements across Europe for (a) and (b) 13 July 1807, (c) and (d) the mean over three days on 1-3 August 1807, and (e and (f) the mean over five days on 26 - 30 August 1807. Grey fills mark stations with less than 25 years of data in the reference period. The highest and lowest three values in the region are labeled.

For all three episodes, 20CR shows a low-pressure system over the East Atlantic / North Sea sector, and ridges of high pressure reaching from the western Mediterranean / Iberia into the continent. South-westerly winds prevail along the (north-) westerly edge of these pressure systems and are associated with large belts of hot air downstream of the ridges. These general features appear in a modulated form for each episode. Specifically, the 13 July episode has the high-pressure ridge in a south-west to north-east direction and it just reaches the Baltic Sea. These analyses suggest that 13 July 1807 was in fact mostly a Central-European event with the largest positive deviations here. On 1-3 August, the more meridional wedge forms into a marked omega high located over the Baltic Sea, with considerably above-normal temperatures between the Czech Lands and Scandinavia, while the episode appears less intense over Western Europe. On 26-30 August, the wedge does not reach as far north and the hot air is again more confined to Western and Central Europe.

Figure 4: Analysis of the 20CR ensemble for (a) a grid point over northern Switzerland (8° E, 47.5 °N) between May and September 1807. Grey lines show 2-meter air temperature (°C) in the realizations from the 80 members, the red line is the ensemble mean. The solid (dashed) black line shows member #3 (#11) with the highest temperature for 13 July 1807, (26 - 30 August period). The panels (b), (c), and (d) show the 20CR fields of 2-meter air temperature (°C; color shades), sea level pressure (hPa; grey solid contour lines), 500-hPa level (decameters; grey dashed contour lines), and wind vectors (850-hPa levels; ms-1). In (b), this is for member #10 and in (c) for member #03 (highest temperature in Stockholm; not shown), and in (d) for member #11.

Overall, we take from the above analyses that the summer of 1807 was indeed extraordinarily hot, especially in terms of three shorter and longer heat episodes. The found circulation patterns during these episodes are similar to the ones that lead to intense warming over the European continent in modern periods (Kautz et al., 2022; Schielicke and Pfahl, 2022; Sousa et al., 2018; Zschenderlein et al., 2020). The synoptic weather dynamics in 20CR are highly consistent with the collected observations and measurements. In all, they reflect a very hot pre-industrial summer, where the measurements may not have reached today's record temperatures at all stations, but where maximum temperature easily reached 30 to 35 °C and more in the Alpine area, for instance, even when considering potential radiation biases in the measurements.






# 5 Performance of the daily reconstruction approaches

The good representation in 20CR makes the summer of 1807 an interesting case to test how well other gridded datasets can represent it. In this section, we evaluate nine such approaches in four aspects of reconstructing the summer of 1807 over time and space.

# 5.1 Temporal aspects: Similarity and discrimination of predicted vs observed temperature and pressure

For the first analyses, the station measurements and the nearest-neighbor values of each dataset are converted to deseasonalized anomalies by subtracting the fit of the first two harmonics (R package geoTS; Tecuapetla-Gomez, 2022) in the case of temperature and the overall mean in the case of pressure, each calculated over the full year of 1807. These can then be compared to measurements from independent stations, although it is hard to derive an overall best dataset from this alone (**Figure S2 in the Supplement**). For a broader view, the visualized statistics in the top rows of **Figure 5** (for temperature) and **Figure S3 in the Supplement** (for pressure) extend the information to all available stations. Hence, the approaches include information from stations that are not independent, but provide a good context in terms of expected average performance or variability among the selected locations. Note here that the spatially best resolved WRF domain is substantially smaller than all other areas and hence the stations from Scandinavia and Iberia cannot be included. Statistics for a congruent spatial extent have also been calculated; differences are small (not shown).

For the extended summer period (May to September 1807), we can roughly discriminate three levels of performance:

- (i) Among the nine approaches, the 20CR ensemble mean (CRM) represents a medium-performance reference point (e.g. mean correlation coefficient of around 0.7, with a large spread). Across all stations, CRB performs slightly better than CRM. The (partial) improvement with respect to CRM comes mostly from more balanced variability (*SDR*).
- (ii) With a few exceptions, the plotting positions of the 20CR ensemble members (including the three members which produced the hottest temperatures for a certain heat episode, cf. **Figure 4**) are detached from the CRM, indicating worse overall performance. In fact, (relative) over-estimation of temperature and pressure in association with potentially lower correlation and higher *cRMSE* can be expected from the nature of 20CR members due to more distinct fields of temperature and pressure.
- For this reason, we refrain from including the 20CR ensemble members in the following analyses. WRF and ONN are often inferior to the competitors in terms of correlation and *cRMSE*, although WRF (ONN) has a more balanced variability than CRM regarding temperature and pressure (pressure only).
- (iii) Rather high-performance statistics come from the remaining approaches CRP, CPB, ARM, TNN, and VAE. We find high correlations (between 0.75 and 0.9), and *cRMSE* statistics mostly below 3 °C (even near 2 °C for Zurich; similar range for pressure [hPa]). Overall, VAE and TNN may be the best performing approach from visual examination of the plotting positions.



Given the distant past and the scarcity of available measurements of temperature and pressure, we argue that the obtained average values speak for the quality of most reconstructions. However, most approaches also show a number of outlier stations with low or very low performance, e.g. at Marschlins.

Figure 5: Beanplots of performance measures and statistics (see Section 3 for abbreviations; arrows mark direction of increasing performance) for temperature (anomalies wrt fit from the combined 1st and 2nd harmonic waves) and nine gridded datasets (color shades; x-axis) and for the period May to September 1807. Black horizontal bars in the beans mark all available values. Values for independent stations are marked with symbols (top-left panel). Black horizontal bars across each bean show mean values, the dotted black horizontal line the mean over all datasets. The stations named at the bottom of each panel most frequently (number after station name) have the highest (left) and lowest (right) statistics across the nine models (min and max absolute distance to 1 for panel sdr); the first example is shown for ties. For graphical reasons, some beans exceed the mathematical limits for some measures and statistics.

To assess the discrimination potential, the bottom panels of **Figure 5** and **Figure S3 in the Supplement** quantify the ability of each approach to accurately predict very hot, warm and cooler air masses and very high, intermediate, or very low pressure for each station (cf. Section 3). For both independent and dependent temperature stations, the relative plotting positions of the approaches look very similar across all three scores. This is true regarding the width of the beans (bulk and average performance) and the length of the beans (i.e., spread of scores across stations), and it is true regarding both temperature and

440

pressure. On average, all three scores are at least 0.5 or better (in both directions of increased performance, e.g., up to 0.75 or more for BACC and GSS, or 0.25 for MBR for temperature). This indicates that, in general, the reconstructions are consistent with observed patterns and would be sufficiently reliable and accurate to be practically useful. Also, the relative ranking of the approaches wrt CRM remains similar to the first analyses: CRM is a reference, medium performing approach regarding our statistics (together with CRB), whereas the ONN and WRF approaches perform less well on average, even if a couple of independent stations may be equally well or better represented than in CRM. On average, the TNN approach performs best here, followed by VAE, and then CRP and ARM. CRP and VAE may be superior when emphasizing the bulk performance (i.e., if larger width and shorter extent of the beans are preferred).

In all, the two temporal assessments reveal that CRM is a good reference and that with modulations, CRP, CPB, and CRB meet the expectation of being enhanced derivates of 20CR. CRP is even in the group with very good statistics, together with the ARM, TNN, and VAE approaches. On the other side, WRF and ONN perform less well. The larger deviations from the station measurements may be explained by the lack of nudging in WRF, and obviously ONN does not incorporate information about the spatial patterns in its reconstruction approach.

## 5.2 Spatial aspects: Regional representation of temperature and pressure

We go back to the Taylor statistics for a more spatial view on performance. This time, we use the Taylor skill score *TSS* to assess the similarity of the nine approaches to station observations of pressure and temperature for a confined summer period in 1807, i.e. the months of July and August (**Figure 6 and Figures S4, S5, and S6 in the Supplement**). The confined period was chosen to avoid a signal of the annual cycle for temperature; hence, the *TSS* is calculated for anomalies wrt the mean of the confined period. Also, the *TSS* has the advantage of summarizing the Taylor statistics in one number (in terms of *COR* and *SDR*; see Section 3), which makes spatial comparisons easier. In general, most approaches show an improvement over CRM on average, in particular at independent stations. Overall, TNN, VAE and CRP are in front of CPB, ARM (less for pressure) and WRF (for pressure), CRM and CRB are almost equal, and ONN and WRF (for temperature) fall behind.

The largest improvements (*TSS* ≥ 0.5) are found around the Baltic Sea, and partly in Northern Italy. No improvements are seen over (North-) Western Europe (France, BeNeLux) and almost all approaches perform less well (*TSS* ≤ 0.3) over Iberia for both temperature and pressure, although VAE (and partly TNN and ARM, less CRP) produce increased skill compared to

CRM for stations like Valencia or Cadiz, for instance. It appears that for both 20CR-based and ML approaches, substantial enhancements come from ingestion of additional information in some areas where CRM has poor station coverage, and some ML-approaches (especially VAE) seem to nudge the spatial patterns more towards ingested measurements than most 20CR-approaches. While the diverging performances of WRF regarding pressure and temperature remain hard to explain, the larger problems of ONN in representing the summer temperature variability can be attributed to the *SDR* that is clearly lower (around 0.6) than for all other approaches (0.85 to 1.2; not shown). In all, the analysis indicates a shift in the influence towards newly

455

ingested stations (wrt CRM) and hence some modulations in the reconstructed atmospheric patterns. In turn, the analysis points also to potential quality issues of the measurements at some stations (see Section 5.3 for this topic).

Figure 6: Taylor skill scores (color shades) for July-August temperature anomalies in nine approaches. Independent stations are marked with a purple circle. Circles with no available model or station values are empty. The inset shows beanplots of the Taylor skill scores similar to Figure 5, mean skill values are added and drawn (red horizontal bar).

Finally, we visually inspect the atmospheric circulation patterns for the 13 July 1807 episode (**Figure 7**, cf. **Figure 4**) and relate them to the *TRI* (and *SD*) values calculated for the temperature and pressure fields.

The 20CR (-based) approaches indicate an Azores High that is connected to a second high over Eastern Europe. This bridge pattern is accompanied by rather weak lows near Iceland and over Iberia, and is associated with considerable temperature and temperature gradients over much of Continental Europe and the Mediterranean. Consistent with written reports (Erdmann and Brönnimann, 2023), CRM shows temperatures of >30 °C in Rome, for instance, but fails to reflect > 30 °C temperatures as

20






measured and reported north of the Alps. In comparison, the pressure centers are (substantially) more pronounced in CRP and CPB, which is in line with more heat reaching from the Maghreb and Spain towards France, Germany and even into Scandinavia (cf. **Figure 4**). CPB nearly matches the observed values in Paris, Berlin, and Zitenice, for instance. The gradual increase in the pressure dipoles, and the more distinct hot areas may be explained by stepping from arguably smoothed fields by averaging the ensemble in the CRM, to adding more observed information in CRP, to selecting a representative member with CRB and CPB. In line with this, increased *TRI* values (and field *SD*) wrt CRM (approx. 3 for temperature and 1.5 for pressure) appear to be more plausible.

In turn, this means that lower *TRI* values may be less plausible. This is the case for ONN and ARM, even though we would expect increased ruggedness from their finer spatial resolution. The ONN fields resemble more a climatological than a weather-scale pattern: ONN shows one prominent blob of high pressure over Europe, centered over the Alps, and a smooth temperature gradient from north to south. The reason for this is that the ONN algorithms attempt to reduce the mean error of the entire spatial field. An area of high pressure over the Alps is also present in ARM, but ARM includes a second high-pressure core over Eastern Europe, and it features a distinct Iceland low. Both ONN and ARM temperature fields lack spatial variability and they hardly match the spatial structure of observed values, particularly in hotter areas. This smooth atmospheric pattern may also explain the regional bias of ONN found in the *TSS* analyses (cf. **Figure 6**).

In comparison, the atmospheric pressure fields in TNN and VAE are much more pronounced, with a distinct low-pressure system over the western Mediterranean. Concurrently, they show hot areas (>28 °C) reaching from the Mediterranean (VAE) to Central Europe (VAE and TNN), and relatively warm areas into Scandinavia. In contrast, their areas over southwestern Europe are relatively cool. All approaches (maybe except for ONN) feature a prominent hot spot over Central Europe, but only CPB and (partly) CRP and VAE seem to reflect the hottest locations according to measurements and reports.

With modulations, the two hot episodes on 1-3 August and on 26-30 August 1807 show similar atmospheric patterns (**Figures S7 and S8 in the Supplement**). Building up on TRI values in pressure and temperature from CRM, CPB (and CRB) propose deep Iceland lows, and areas of >30 °C in Scandinavia for the first episode, which aligns with observations in this region. They show very hot areas (>30 °C) in much of the southern European sector for the second episode, and areas of still high temperature towards the Baltic Sea. The pressure fields in TNN, VAE and ARM are similar, but none of these produce hot spots on larger scales such as the 20CR-derived reconstructions.



Figure 7: Maps of air temperature (°C; color shade) and sea level pressure (hPa; contour lines) for 13 July 1807 according to nine reconstructions. Reported measurements of temperature are given in colored half circles (left side). Locations of maximum and minimum pressure are marked with purple dots, 1016-hPa (28-°C) contours are highlighted in blue (red). Relative TRI (boxplots) and field SD (red and blue horizontal bars) of air temperature (ta) and sea level pressure (p) are given in bottom left panels.

Some of the atmospheric features may only become evident on a smaller scale, in particular for WRF. **Figure 8** encompasses the area of the WRF domain 02 for July 13, 1807, arguably the hottest summer day in the domain with maximum temperatures of well over 30 °C, for example in Aarau (36.2 °C), Schaffhausen (34.6 °C) or Delémont (33.8 °C). Clearly, only WRF reproduces such heat intensity, not only on the northern side of the Alps, but also for regions such as Paris, Berlin or Prague. The spatial structures also reflect the refinement of the original information from CRM to WRF, and CRB and CPB also look similar. Qualitatively, TNN and VAE and ARM show higher temperatures north of the Alps, and all of them seemingly aim

to capture the measured temperatures in space, most strongly with TNN. Rudimentarily, the pressure fields may even suggest 495 a south foehn situation from a slight pressure gradient over the Alps, which may offer an explanation for the heat that is somewhat different from the circulation suggested by the 20CR-based approaches. The spatial pattern of the WRF refinement with plausible heat intensity and the arguably increased nudging towards station measurements of the non-20CR approaches (except ONN) is also evident in the other two episodes (not shown).



In all, the analyses add to the plausibility of the 20CR (-based) output of atmospheric temperature and pressure fields. The 20CR-based approaches appear to have a clear benefit from the flow awareness – the atmospheric circulation can easily be explained from the produced fields. Specifically, the assimilation of additional measurements, as well as the selection of a best ensemble member, enhance the local representation of the fields and is reflected in the generally increasing 'ruggedness' towards CRP and CPB compared to CRM. This enhancement in spatial patterns, however, comes at the cost of lower TSS scores at other locations. This again points to increased influence of the additionally ingested station measurements. Also, the fact that WRF only refines information from CRM, not from additional stations, explains the rather large discrepancies in temperature values at some places, and hence the low statistical performance in the previous sections.

VAE, and partly TNN, propose different atmospheric patterns that may be plausible for the core areas of the study. However, 510 they do not show extremely hot conditions over Western Europe, in particular the Iberian Peninsula, which points to a differing handling of the available measurements, i.e. to a potential over-confidence in erroneous measurements. The TNN, and particularly the ONN approaches appear to have problems in plausibly representing regional-scale atmospheric variability during the hot episodes. ARM takes a position between these two groups of approaches: Qualitatively, the ARM fields tend to remain quite close to the (well-represented) stations. It maintains plausible atmospheric patterns, although at the cost of intensity at some instances.


Figure 8: As in Figure 7, but for the WRF domain 02 only, and using every 2<sup>nd</sup> CRM grid point (blue crosses in top-left panel) for calculating the terrain ruggedness index and the standard deviation over the field (colored horizontal bars in the boxplots).

## 5.3 Quality of the station measurements



All available stations (our 'ground truth') have been included in the above analyses, irrespective of the arguably varying quality of the measurements. Clearly, some stations are notoriously among the best or worst performing in the applied measures and statistics (**Figure 5**, **Figure 6**, **Figure S3**, **and Figure S4** in the **Supplement**). In fact, we know from station records such as Marschlins that it contains suspicious values, e.g. some strong changes from one measurement to the other, or implausibly







high temperature values at some instances. In contrast, the stations of Barcelona, Valencia and Cadiz have many values that only differ minimally over time. Still, we refrained from excluding these stations from the analyses. One reason is that including them lets us see how the approaches deal with potentially erroneous measurements. Another reason is that we do not have the means to objectively check all observations with regards to modern data, to an enlarged set of climatological data, or even to neighboring stations. In turn, we assume that systematically strong differences in statistical performance wrt other stations point to lower quality.

For illustration, Figure 9 and Figure S9 in the Supplement show a selection of arguably high-quality versus low-quality stations. The discrimination comes from a semi-subjective assessment, where acceptable to good values in the Taylor diagrams are defined as bulk COR values for temperature of 0.4 to 0.6 (0.5 and 0.7 for pressure) or above, cRMSE of 3° C or 2° C or below for temperature (5 hPa and 3 hPa for pressure). For SDR, factors between approximately 0.5 and 2 are acceptable, and factors between 0.75 and 1.5 are good. Note that only mid- to high-performing approaches are considered here (i.e. all except for WRF and ONN). The shown low-quality stations hardly fulfill any of the criteria, whereas the high-quality stations mostly fulfill the more rigorous criteria. Such high-quality stations, among others, are Milan, Prague, Delémont for temperature, or Rovereto, Karlsruhe, and Schaffhausen for pressure. Note also that in this context, our independent stations appear to be of rather high quality.

Specific features of low quality are very low correlations and variability (COR; SDR; e.g., Valencia), implausible deflections in the anomalies (cRMSE; Marschlins or Valencia), or phases shifted on the time axis in the simulations versus observations (Valencia, Marschlins). Not surprisingly, these are also the stations with the lowest TSS in Figure 6 and Figure S4 in the Supplement. However, too low variability could also be a problem of the reconstruction approaches, especially 20CR. Stockholm (possibly also St. Petersburg) is such an example, where the anomalies are not a priori implausible. Rather, it seems that the 20CR-based approaches in particular have problems here. 20CR has little station data available for assimilation in this region, and as a consequence, the results become clearly better with additional assimilation (St. Petersburg, Stockholm in parts).

In all, the 20CR (-enhanced) approaches may suffer in areas with no or hardly any stations, in addition to the fact that the assimilation algorithms exclude measurements that are far off expectations (cf. Section 2.2.2). In turn, the more data-driven approaches like TNN and ONN (and partly ARM) are not able to exclude or downweigh low-quality measurements. For instance, they simulate much lower temperature over Iberia compared to 20CR, or produce very pronounced low pressure over the same region (TNN in particular), which are harder to explain in the context of a heatwave than the 20CR-based fields (cf. Section 4). This means that adding even a small number of high-quality measurement series, ideally from less-covered regions, leads to valuable improvements for any considered approach.


Figure 9: Taylor diagrams (top area in each panel), time series (bottom left area in each panel), and climatology baseline (bottom right area of each panel; fit from the combined first two harmonic waves for temperature) of observed (black circle or black bold line) versus predicted (color dots or dashed lines; see image legend for abbreviations) anomalies of 2-meter air temperature (ta) for a selection of three high-quality (top row) and low-quality stations (bottom row) and the period May to September 1807.






#### 6 Summary and conclusions

Recent advances in producing gridded datasets have addressed the lack of comprehensive weather information before around 1850. Efforts to recover and integrate early instrumental pressure measurements have supported the development of global atmospheric reanalyses such as the 20CR. More recent approaches can also be used to generate daily weather fields from early weather records in (Central) Europe and the greater Alpine region, encompassing (i) 20CR-derived approaches, (ii) analog resampling methods, and (iii) ML algorithms. This study aims to assess nine such approaches in reproducing the daily weather during the extremely hot, pre-industrial summer of 1807 in Europe. The datasets include the 20CR ensemble mean (CRM) and enhanced versions from 20CR by using assimilation of so far unused station measurements of temperature and pressure (CRP), the best raw analog member (CRB), and the best analog member after assimilation (CPB). The WRF dataset is built by dynamical downscaling from CRM using nested weather model domains. The analog reconstruction method (ARM) searches for similar weather patterns in a modern reanalysis product. Three numerical datasets are based on machine-learning approaches, such as one-dimensional neural networks (ONN), a three-dimensional completion method as in video inpainting (TNN), and a variational auto-encoder (VAE) to isolate the essential features.

The collected evidence from eye observations, early instrumental measurements, and from CRM confirms that the summer of 1807 in Europe was indeed extremely hot and can be considered a prototype heatwave summer within a pre-industrial context. We find specifically hot episodes on 13 July (with local temperature measurements north of the Alps of around 35  $^{\circ}$ C), on 1 – 3 August (in Scandinavia / North-Eastern Europe), and on 26 – 30 August (in Central and Western Europe).

We use the Taylor diagram measures (i.e., *COR*, *SDR*, and *cRMSE*) to assess the similarity of predicted to observed temperature and pressure values. We find very good correlations for most data sets (> 0.75 for temperature and > 0.5 for pressure anomalies), cRMSE of less than 3 °C for temperature (similar range in hPa for pressure), and a tendency to underestimate variability. These numbers speak for the quality of most reconstructions, especially given the distant past and the scarcity of underlying weather information. We find that the 20CR mean (CRM) represents a mid-performance reference point, whereas most 20CR ensemble members as well as the WRF and ONN approaches do not perform as well. On the other

end, TNN, VAE, CRP, and ARM are often among the best performers; see also **Figure 10** for a (semi-subjective) visual summary of the relative performances.


Figure 10: Summary ranking (y-axis, top is best) for the four aspects of performance (grey shades) and for the nine considered datasets, assessed with regards to CRM as a reference. Darkest grey are the averaged ranking points for the Taylor measures, i.e. in the aspect of similarity to (independent) station measurements of temperature and pressure (anomalies wrt station means), dark grey are the averaged ranking points for the discrimination statistics (extreme versus more normal conditions), light grey are the ranks for spatial improvement using the Taylor Skill Score and subjective assessments, and lightest grey are the spatial circulation patterns, assessed for TRI and (subjective) dynamical plausibility of temperature and pressure fields.



Next, we use the *BACC*, *MBR*, and *GSS* scores to see how well the approaches can discriminate between hot, warm, and cooler conditions (very high, intermediate, and very low pressure). On average, all three scores are at least 0.5 or better (e.g. up to 0.75 or more; extreme scores are near 0 and 1) for almost all approaches, which means that the reconstructions are consistent with observed patterns. Again, CRM (and CRB) can be seen as good reference datasets. Most data-driven ML datasets perform very well here. The TNN approach is generally best, followed closely by the VAE, ARM and CRP approaches.


Then, we us the TSS to assess the spatio-temporal skill. No approach performs well in all regions. Improvements ( $TSS \ge 0.5$ ) primarily occur over Central / Northeastern Europe, while Western Europe shows equal to lower skill compared to CRM ( $TSS \le 0.3$ ). We conclude that the atmospheric patterns in the reconstructions are influenced by this shift in weight towards regions with substantial (new) information; best scores come from reconstructions that are nudged towards (additional) observations, such as the ML approaches (TNN, VAE), and ARM, but also CRP in parts.

In a final analysis of the heat episodes, the 20CR-based approaches feature modulated forms of an Azores High and a ridge across the Continent, and they (almost) reproduce measurements of up to 30 °C or more, e.g. at places like Paris, Berlin, or Zitenice. This aligns with known atmospheric patterns of hot air reaching from South-Western towards Central Europe and Scandinavia. Uniquely, the WRF refinements capture regional temperature maxima north of the Alps. In contrast to the 20CRbased datasets, ARM, TNN and VAE feature a (distinct) low-pressure system and lower temperature over Iberia. However, the generated atmospheric patterns seem less consistent with known large-scale heatwaves.


We conclude from the analyses that the atmospheric flow-aware datasets clearly benefit from a physics-constraint approach, providing easily explainable atmospheric circulation patterns. Enhancements are linked to stepping from potential smoothing effects in CRM to more assimilations in CRP and more distinct realizations (CRB, CPB) and to spatial refinements in WRF. However, the latter reconstructions also tend to run (too) freely from original constraints at places, and are therefore often penalized in statistical scores.

The data-driven ML approaches, on the other hand, excel in the statistics. TNN seems the most powerful approach for predicting local station values, closely followed by VAE. However, both struggle with plausibly representing regional-scale atmospheric circulation. Partly, this may be explained by over-confidence regarding potentially erroneous, inconsistent or uncertain observations. In particular, ONN strives to reduce the mean spatial field error, which results in loss of regional variance and skill, but its strengths lie in the extremely rapid, low-resource assessment of the atmospheric situation.

The ARM approach stands in a middle-ground, it seeks not to be too far to station measurements, which is reflected in good statistics. In turn, the conservative approach leads to low-intensity weather patterns.

Overall, no single approach appears on top of all others; most have excellent scores in one field at the cost of poorer scores in

another. This means that future users and developers may select and weight the properties that are important to them from various dimensions; e.g., cost-effective versus sophisticated configurations, atmospheric flow-aware versus data-driven approaches, more local versus more large-scale performance. There is potential in many dimensions. It is apparent that ingestion of additional records helps, but further developments need to address the uncertainties and potential errors in the early measurements. In turn, this also means that the quality of input weather records remains crucial, and emphasizes the

value of metadata, quality checks, unit conversions or homogenization, where applicable.

\*\*\*\*\*\*\*

# Plain language summary

We test nine reconstructions of Europe's hot summer of 1807, using weather records, reanalyses, machine-learning (ML), and data assimilation. Most approaches match observed temperature and pressure well. Approaches with

explicit information about atmospheric flow capture weather patterns well, while ML approaches better reflect local measurements. Ingestion of accurate records from new regions improves the reconstructions markedly. In all, the approaches provide new insights to pre-industrial extreme weather.

# Data and code availability

Pressure measurements from the International Surface Pressure Databank Version 4.7 (ISPD) are available at 645 https://doi.org/10.5065/9EYR-TY90 (Compo et al., 2019). Station measurements related to the CHIMES project are available **PANGAEA** https://doi.org/10.1594/PANGAEA.948258 from (Brugnara, 2022) and https://doi.org/10.1594/PANGAEA.961277 (Brugnara et al., 2023). The 20th Century Reanalysis version 3 (Slivinski et al., 2019) ensemble members back to 1836 can be downloaded from NERSC at https://portal.nersc.gov/project/20C Reanalysis/ (National Energy Research Scientific Computing Center, 2019). The experimental extension of 20CR back to 1806 is 650 obtainable from NOAA upon request (psl.data@noaa.gov). Wrapper scripts and other tools used for running WRF and WRFDA are available via the NCAR WRFDA users' page https://www2.mmm.ucar.edu/wrf/users/wrfda/download/tools.html (National Center for Atmospheric Research, 2024). R software code is available from the packages mentioned in the main text. The used datasets are stored on the University of Bern Open Repository and Information System, BORIS (Stucki, 2025; https://doi.org/10.48620/91594). The VAE codes are available from https://github.com/conallruth/Weather-Reconstruction-655 VAE.

## **Author contributions**

SB and PS planned the campaign.

YB, JF, RP, JF, LP, NI, PS and SB collected and pre-processed the observational data.

SB, NI, LP, YS, MW, and CR produced and provided the gridded datasets.

PS analyzed the data, prepared all visualizations and wrote the article.

SB, NI, MW, CR, RP, JF and YB reviewed the manuscript.

## **Competing interests**

The authors declare that they have no conflict of interest.

# Acknowledgements

Support for the Twentieth Century Reanalysis Project version 3 dataset is provided by the U.S. Department of Energy, Office of Science Biological and Environmental Research (BER), by the National Oceanic and Atmospheric Administration Climate Program Office, and by the NOAA Physical Sciences Laboratory. We like to thank numerous digitizers that contributed to make the historical data available. We used freely available AI-powered language models such as GPT, Claude, Llama, Mistral, DeepL, or Perplexity for occasional checks of vocabulary, grammar, style, and structure of chapters and paragraphs.

## 670 Financial support

Funding for LP, PS, and YB came from the Swiss Early Instrumental Meteorological Data (CHIMES) and the Long Swiss Meteorological Series projects; the latter was funded by the Global Climate Observing System (GCOS) in Switzerland. Funding for NI, LP, and PS came also from the Daily Weather Reconstructions to Study Decadal Climate Swings (WeaR) project (Swiss National Science Foundation, grant number 188701).

Funding for CR, NI and PS came also from the Decadal Variability of Daily Weather (DVDW) project (Swiss National Science Foundation grant number 219746). YB has also been funded by the Copernicus Climate Change Services (C3S) 311c Lots 1 and 2.

The research work of JF and RP was supported by a grant entitled 'The occurrence of extreme weather, climate and water events in Poland from the 11th to 18th centuries in the light of multiproxy data', funded by the National Science Centre, Poland (Grant No 2020/37/B/ST10/00710).

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
