# Peer review of "Evaluation of nine gridded daily weather reconstructions for the European heatwave summer of 1807"

_EGUsphere, 2025_

## Author Comment (AC1)

**Comment by Rhidian Thomas**

https://egusphere.copernicus.org/preprints/2025/egusphere-2025-5264/#discussion

**RC1**: 'Comment on egusphere-2025-5264', Rhidian Thomas, 21 Nov 2025 reply

This manuscript presents a detailed and useful study of an important event: the summer of 1807 in Central Europe, a "prototype heatwave summer within a pre-industrial context". The study's aim is to critically compare nine different gridded reconstructions for this event, comprising:

- Four methods derived directly from the 20CRv3 reanalysis, either in its original form or with offline assimilation of additional observations;

- A downscaled simulation using 20CR fields as boundary conditions;

- An analogue resampling method trained on ERA5 fields;

- Three machine learning (ML) methods of varying complexity.

The authors identify hot periods in the 1807 summer using a variety of data sources: qualitative accounts from a contemporary observer, daily station temperature series, and reconstructions using the 20CR ensemble. The bulk of the paper is given to a thorough evaluation of the nine methods, using both statistical criteria (Taylor diagrams and three extreme-specific metrics) and a semi-quantitative analysis of the spatial plausibility of the reconstructed fields. Finally, the authors critically evaluate the quality of certain stations and consider how this will affect each method differently.

Various approaches are now emerging for reconstructing historical weather, and the authors cite these in the introduction. The authors show that several of these methods can produce plausible reconstructions of the 1807 summer, a well-chosen case study that is of increasing significance in the current changing climate. The manuscript also provides a rigorous framework for comparing reconstruction methods that will be useful beyond this single case study. Overall, the authors present an impressive amount of information that will be of broad interest to readers of Climate of the Past. A highlight of the paper is the novel inclusion of reconstructions using a variational auto-encoder (VAE), and the authors find interesting differences between VAE and the other ML approaches compared to the physics-based reconstructions.

The paper is well organised and well written, and the figures are clear and helpful. The title is accurate and the abstract gives a good overview of the findings. I have two general comments and a few science comments below which I feel would improve the manuscript, however these are fairly minor and I do not feel they amount to major edits. Otherwise, I am pleased to recommend it for publication in Climate of the Past.

The comments below are organised into two general comments, several specific science comments, and technical/typing comments.

**Ensemble-means vs best members**

In L379, the authors state that "CRB performs slightly better than CRM". My interpretation of the results was the opposite – I thought it was interesting that the best member datasets (CRB and CPB) generally performed no better than their corresponding ensemble means (CRM and CRP respectively). I think L379 refers to Figs 5 and S3; here, CRM has slightly lower COR for ta, but higher for p. It's also not clear that CRB has "more balanced variability" – this may be true for p, but for ta the SDR looks closer to 1 in CRM than in CRB. I interpreted Figs 6 and S4-S6 similarly, where the TSS scores show little improvement in the best members compared to ensemble means (CPB in S5 is an exception here). This is not a huge point, but I think it detracts from an important conclusion that the authors draw (e.g. L586 in the summary): that CRM is a good "mid-performance reference point" that is not easily beaten even by concatenating the best individual ensemble members. As 20CR is such a widely used dataset, I think it is important to highlight this result for other users.

Thank you for this closer look. It is indeed an important detail that has implications for the final interpretation. From visual comparison of the three measures in the top rows of Figs 5 and S3, we find the following superiorities: COR: ta CRM, p CRB; cRMSE: ta, p CRM; SDR: ta CRM, p CRB. We will hence rephrase to:
"Across all stations, CRM performs even slightly better than CRB. The (partial) improvement with respect to CRB comes mostly from overall better cRMSE values."

In the summary, we will adopt your suggestions and write that CRM is a good "mid-performance reference point that is not easily beaten even by concatenating the best individual 20CR ensemble members"

**Temporal evolution**

The flow fields in Fig 7 and Figs S7-S8 show the reconstructed circulation at specific snapshots in time (single day for Fig 7, a few days average for S7-S8). But if we were interested in development of a system over time, we would want the fields to change smoothly from day t to day t+1. Do the ML approaches show this property, or do you occasionally see unrealistic "jumps" between days? For example, if there are multiple local minima (circulation patterns) that the model could end up in, it could conceivably settle in different minima on successive days as the fitting is done separately for each day. This does not appear to occur in ML weather models in the present day, but I wondered if it is more of an issue for historical periods due to the sparsity of input observations – I would guess that, as the observational constraints become weaker, there are more possible circulation patterns that could fit the input at each timestep. In the VAE approach, for example, is the model constrained in any way to produce fields that are smooth in time?

I am not asking for any extra work to address this query – I am just interested to see if the authors noticed any differences between the methods here, or if I have misunderstood some aspect of the ML methods. If they noticed interesting differences, it might be a nice addition to their discussion of the strengths of each method.

Thank you for pointing this out, as indeed, some readers might be interested in the jumpiness of our approaches. It has motivated us to perform a few more analyses. In the end, we suggest that we introduce the text below as a short summary and interpretation (after line 500), but do not include the boxplots added below for your interest.

"So far, our focus has been on single daily fields or multi-day composites. However, some readers may be interested in the day-to-day evolution of atmospheric circulation. For example, the pressure field over Europe should evolve smoothly from one day to the next, in most cases. To assess whether our approaches introduce large jumps, we calculate the root mean square deviation, mean absolute deviation (MAD), and Pearson correlation coefficient between consecutive days (t and t+1) for the ARM field extent. All fields are interpolated to a 0.5° grid.

All metrics show similar patterns of smoothness or jumpiness (not shown). The average day-to-day MAD for 2018 is highest for the best-member approaches (CRB and CPB), at approximately 6.5 hPa, followed by ARM at about 6.0 hPa. Approaches using a "best fit per day" can therefore produce rather jumpy pressure field evolutions. The lowest day-to-day differences (2–3 hPa) occur for CRM and CRP, likely due to averaging and excluding outlying observations. The ML approaches TNN and VAE fall in between, at about 4–5 hPa. They lack past-day memory or built-in smoothing, but rely on the local evolution of observations."

[Figure]

*Figure 1: Metrics of pairwise (day t to day t+1) evolution of the pressure fields over a European Sector (ARM domain), for seven approaches. Shown are the distributions of day-to-day field means across the year of 1807.*

L181: What are the two periods used to calculate the temperature offset? Is the past period a single year (1807) or an average? The difference could be sensitive to the start year, so averaging over a period (e.g. 1800-1810) may be best.

The past period is a 50-year window centered on 1807, while the present-day period spans 1951–2003, covering the ERA5 and EKF400 periods. The resulting difference is therefore not sensitive to the exact choice of start year. The sentence in the manuscript will be adapted to: "This offset is defined as the difference in zonal-mean, land-only temperature from the EKF400 palaeo-reanalysis (Valler et al., 2021) between a 50-year period centered on 1807 and a present-day period from 1951 to 2003, as represented by the ERA5 and EKF400 datasets."

L381-384: "With a few exceptions, the plotting positions of the 20CR ensemble members (including the three members which produced the hottest temperatures for a certain heat episode, cf. Figure 4) are detached from the CRM" – I don't think the 20CR ensemble members (x80) are shown in Fig 5 or Fig S3? Do you mean the best-members methods (CRB and CPB)?

There is a wrong reference, thank you for the hint. We will replace 'cf. Figure 4' with 'cf. Figure 2 in the Supplement'.

"In fact, (relative) over-estimation of temperature and pressure in association with potentially lower correlation and higher cRMSE can be expected from the nature of 20CR members due to more distinct fields of temperature and pressure" What do you mean by "more distinct fields of temperature and pressure"? More distinct than what (the ensemble mean)?

Yes, we will complete the sentence to: 'more distinct fields of temperature and pressure than the ensemble mean'.

L410: I was unsure how to interpret this sentence. Does it mean the average *across all methods* is better than 0.5 for each of the three scores? I interpret this to mean the dashed line is better than 0.5 (higher or lower depending on the score). But then what do the values of 0.25 and 0.75 refer to? Also, do the values in this sentence refer only to ta in Fig 5 (the values for p in S3 seem different)?

I think we just used a wrong and misleading wording here, we will replace 'On average' with 'In most cases'.

L431: I think this is a helpful summary of the performance of each method. It looks like the ordering follows the ordering of the TSS score in the lower right panels of Figs 6 and S4-6 – if so, it might be helpful to state that, e.g.: "Overall, the methods can be ranked by their TSS scores: TNN and VAE…." etc.

Thank you for this simple and very clarifying sentence, we will adopt it.

L20 (Table 1): What does "Time" mean here – e.g. does 14 mean 14:00 UTC? Could clarify in the caption.

Thank you for the hint: Here, 14 means 14:00 Local Mean Time, we will clarify in the caption: "Time indications like 14 refer to local time, i.e. 14:00 LMT."

L191: I'm not sure what the end of this sentence means – missing a word?

We will change the end of this sentence to "designed for weather reconstruction".

L268: Does [0,1] here mean any value between 0 and 1? It may help to say this in words as well.

Yes, this might be misleading, we will replace it with "(0 or 1 in our case)".

L269: This sounds like the best performance is when MBR=2 – I think it should be when MBR=0?

Thank you for this observation — we agree that the wording leads to confusion and appreciate the chance to clarify.

MBR is computed using the unnormalized formulation (Brier 1950; as also used e.g. in mlr3measures::mbrier), where the score ranges from 0 (perfect prediction) to 2 (worst-case, i.e. constantly assigning probability 1 to an incorrect class). Lower values therefore indicate better performance, with the optimal value being MBR = 0.

We did not intend to suggest that MBR = 2 represents the best performance; the statement "for best performance" in the text was meant to refer to the lower end of the range (0). To avoid any ambiguity, we will revise the sentence in question as follows:

" the range of the 3 x 3 $MBR$ goes from 0 for perfect predictions to 2 for worst-case predictions, i.e. lower values indicating better performance".

L282: It allows *us* to summarize

We will adopt the suggested change.

L297: I would possibly avoid using "tendency" here, due to its other meaning (d/dt) which could be confusing. You could just end the sentence after "appear more rugged".

We will adopt the suggested change.

L604: us --> use

We will adopt the suggested change.

Fig 2: Units are missing for the y-axis – can add these either in the figure or in the caption.

Thank you for the close look. We will change the caption to "Daily mean temperatures (°C; y-axes) at Prague,..."

---

## Author Comment (AC2)

**Comment by Philip Gooding**

https://egusphere.copernicus.org/preprints/2025/egusphere-2025-5264/#discussion

RC2: 'Comment on egusphere-2025-5264', Philip Gooding, 15 Dec 2025 reply

Due to my training and expertise, my comments focus largely on the use of weather records. I note the first reviewer has provided more substantive comments on the interpretation of different ML and reanalysis data - I don't have anything to add here. I do, however, second their view that 'the authors present an impressive amount of information that will be of broad interest to readers of Climate of the Past.'

L525. I wonder if the authors can do more to address the issue of 'suspicious values' in weather records. Their approach - leaving them in to 'see how the approaches deal with potentially erroneous measurements' and because they 'do not have the means to objectively check all observations' - is certainly valid. However, it doesn't necessarily have to be a case of 'inclusion' or 'exclusion' of suspicious values. Have the authors considered developing a confidence scale, thereby indicating in quantitative terms which sets of weather records they a more confident in? They allude to certain datasets as being probably of higher quality (e.g. Barcelona) than others (e.g. Marschlins). Would quantifying such 'quality' (and displaying it in the form of a table?) enhance the analysis?

We were unsure about including our assessment of confidence in the station measurements. We refrained because it was done ex post facto, using semi-quantitative methods without state-of-the-art techniques such as homogenization. However, we recognize that the information could be valuable to some readers and will include a column in Table 1 presenting the outcome of the assessment numerically: 2 means that we have good confidence, 1 means medium, and 0 means poor confidence.

L632. 'There is potential in many dimensions.' This is true. However, could the authors be more specific? One of the strengths of this manuscript is that it incorporates data and perspectives from diverse sources and methods. Many readers (including myself) may only be a specialist in one of these fields. Thus, I wonder if the authors could speak to specific avenues for research that such specialists could undertake (apart from, of course, building more interdisciplinary collaborations)? What could a specialist in old weather records/reanalysis/ML take from your article, and what could they contribute to your interdisciplinary efforts moving forward? What questions for further research does your manuscript bring up for specialists in different fields?

Thank you for this comment. It has made us think more about the roles you mention, and we may change the last paragraph of the manuscript into a slightly longer explanation, such as

"There is potential in many dimensions. Specialists in historical weather observations and measurements can help by providing accurate data. Records from areas with sparse

coverage—such as the North Atlantic in our analyses—are particularly helpful. In turn, this also highlights the value of metadata, quality checks, unit conversions or homogenization. Regional reanalysis experts may contribute by developing frameworks that can include more potentially valuable measurements. While reanalyses often discard values that deviate too strongly from expectations, such outliers sometimes reflect actual weather conditions and thus hold significant value. Machine learning approaches, in turn, must learn to cope with the uncertainties inherent in early measurements, e.g. by designing models that can assess potential errors without discarding useful information. Overall, this calls for collaboration between specialists in historical data recovery, reanalysis, and machine learning—each addressing complementary aspects of data quality, uncertainty, and model representation."

The clarity of figure 10 would be enhanced if it were in colour (as in the other figures).

We had tried a few color schemes. However, we felt that they either visually over-represent one column (e.g. brighter colors, or orange, etc.) or could lead to confusion with similar color assignments to other variables in the manuscript. Finally, we used the grey scale because of 'visual equidistance' of the bars.